# Regulating Pathways of *Bacillus pumilus* Adamalysin-like Metalloendopeptidase Expression

**DOI:** 10.3390/ijms25010062

**Published:** 2023-12-20

**Authors:** Natalia L. Rudakova, Albina R. Sabirova, Damir I. Khasanov, Iuliia V. Danilova, Margarita R. Sharipova

**Affiliations:** Institute of Fundamental Medicine, Kazan Federal University, Kremlevskaya St. 18, 420008 Kazan, Russia; natalialrudakova@mail.ru (N.L.R.); hasda2149@gmail.com (D.I.K.); danilova146@mail.ru (I.V.D.)

**Keywords:** Spo0A, DegU, signal transduction system, sporulation, regulatory mutants

## Abstract

The minor secreted proteinase of *B. pumilus* 3-19 MprBp classified as the unique bacillary adamalysin-like enzyme of the metzincin clan. The functional role of this metalloproteinase in the bacilli cells is not clear. Analysis of the regulatory region of the *mprBp* gene showed the presence of potential binding sites to the transcription regulatory factors Spo0A (sporulation) and DegU (biodegradation). The study of *mprBp* activity in mutant strains of *B. subtilis* defective in regulatory proteins of the Spo- and Deg-systems showed that the *mprBp* gene is partially controlled by the Deg-system of signal transduction and independent from the Spo-system.

## 1. Introduction

*Bacillus pumilus* strain 3-19 is a derivative of the *Bacillus pumilus* 7P soil isolate with acquired resistance to streptomycin, and with increased expression of some extracellular hydrolases [1]. The metalloendopeptidase MprBp is secreted into the culture medium by *Bacillus pumilus* 3-19 cells at the end of the exponential growth phase and acts as a minor protein. The protein was purified to homogeneity, sequenced by mass spectroscopy, and biochemically characterized [2]. Based on the primary structure of the conserved motifs of the active centre and Met-turn the enzyme has been classified as a metalloproteinase belonging to the metzincin clan. It combines features of two families, the astacins and adamalysins/reprolysins. Notably, this represents the first prokaryotic homologue of the eukaryotic adamalysins/reprolysins family [3]. Metalloproteinases from this family have not previously been described for bacilli. Adamalysins perform protective and regulatory functions in eukaryotic cells, controlling physiological and pathological processes in living organisms. The functional role of the bacillary homologue is unclear due to its extremely low content in the medium. Elucidation of the regulatory networks controlling metalloproteinase activity will enable the evaluation of this enzyme’s contribution to the integrated cellular response.

Gene expression can be assessed directly through RT-qPCR, which is considered the most reliable and informative method [4,5]. However, gene expression can also be determined indirectly by evaluating the activity level of its product [6,7]. For this study, the level of proteolytic activity of metalloendopeptidase was utilized as an indirect parameter for assessing the expression of the *mprBp* gene. It has previously been established that the mechanisms of carbon and nitrogen catabolite repression regulate the expression of the metalloendopeptidase gene. The synthesis of metalloproteinase is blocked by excess glucose in the culture medium. In experiments involving strains mutant for nitrogen metabolism regulatory proteins, it was demonstrated that the ammonium transport proteins GlnK and AmtB, which interact with the TnrA regulator, play a role in controlling the activity of metalloendopeptidase [8].

Minor extracellular proteinases of bacilli may be involved in the mechanisms driving cultural survival strategy changes, such as the transition from free-floating cells to biofilm formation [9,10]. Numerous studies indicate a strong link between DegU and Spo0A regulators in adjustment procedures while bacilli transit to spore and biofilm formation [11,12,13,14,15]. In addition, adaptive properties such as mobility, the production of antimicrobial molecules and the ability to colonise plant roots may be positively or negatively regulated by the cooperation between DegS/DegU and Spo0A [16].

The aim of this study is to elucidate the role of the Deg and Spo signalling systems in regulating the activity of the *Bacillus pumilus* 3-19 metalloendopeptidase.

## 2. Results

### 2.1. Promoter Region Analysis of mprBp

The full genome sequencing data for *B. pumilus* strain 3-19 facilitated the study of the *mprBp* metalloproteinase gene (GenBank: ACE75740.2) and its promoter region, leading to the selection of primers for efficient cloning of the gene [1]. The gene sequence under its own promoter, spanning 1.1 kb in total, was then inserted into the expression vector pCB22. The newly formed vector containing the metalloproteinase gene was designated as pSA1.

In order to assess the potential control of the metalloproteinase activity, the regulatory region of the *mprBp* gene was analysed for conservative binding sites with regulatory proteins. The 400-nucleotide promoter region was aligned with canonical sequences to allow for interaction with the Spo0A transcription factor, which is involved in spore formation initiation, and the DegU transcription factor, which controls the expression of biodegradation proteins. Genetic analysis identified binding sites within the *mprBp* gene promoter. The degree of homology to the consensus is shown in Table 1 and Figure 1.

Possible binding sites in the promoter region of *mprBp* for the regulatory proteins DegU and Spo0A suggested that the expression of the gene is controlled by these systems. In order to test this hypothesis, the metalloproteinase gene was cloned into the *Bacillus* strains lacking the genes for the regular proteins mentioned above.

### 2.2. Effect of DegU Transcription Factor on Metalloendopeptidase Activity

The *mprBp* metalloproteinase gene was cloned into several *B. subtilis* strains with mutations in the genes of the DegS-DegU regulatory system. The study of MprBp activity in *B. subtilis* strain 8g5Δ*degS*Δ*degU* pSA1, which lacks the DegS and DegU regulatory proteins, showed an average reduction of 80% in the productivity of metalloproteinase synthesis compared to *B. subtilis* strain 8g5 pSA1 with a fully functional DegS-DegU system (Figure 2). Nonetheless, the lack of the regulatory pair did not fully suppress *mprBp* gene. Therefore, while the Deg-system plays a role in controlling proteinase synthesis, it is not the sole regulator of *mprBp* gene function.

MprBp activity was studied in the mutant *B. subtilis* strain 8g5 degU32(Hy), which has a mutation in the *degU* gene, responsible for the stabilisation of the phosphorylated form of DegU~P protein [19]. This mutation is known to cause a significant increase in the expression of genes positively regulated by the DegS-DegU system. The data indicate a tenfold increase in the metalloproteinase productivity of the recombinant *B. subtilis* strain 8g5DegU32 (Hy) pSA1 (Figure 2). We concluded that it is the phosphorylated form of the DegU protein that stimulates the activity of MprBp. The DegS-DegU regulatory pair facilitates the activity of the metalloproteinase, which contributes to the cell’s adaptation during the transition to the stationary growth phase wherein numerous signal transduction systems are triggered [20].

### 2.3. Effect of Spo- Regulatory Proteins on MprBp Activity

The metalloproteinase gene was cloned into several strains with deleted genes of Spo-system regulatory proteins, derived from *B. subtilis* strain 168 (trpC2). It was found that the productivity of the metalloendopeptidase in the recombinant strain with a deficient Spo0A regulatory protein was maintained at the level of the strain with the complete Spo0A gene (Figure 3). The MprBp productivity level showed consistent results in strains with defective spore-specific regulatory proteins, namely Spo0B, Spo0F, Spo0K, and Spo0J, and SigF, SigH, and SigK. None of the spo-regulated mutants showed an altered activity level of the MprBp compared to the control strain *B. subtilis* 168 (trpC2) that expressed the full complement of the corresponding protein. These results suggest that the *mprBp* gene is independent of Spo-regulatory proteins.

## 3. Discussion

The results obtained reliably demonstrate that the DegS-DegU signalling system subcontrols the work of the metalloproteinase and its independence from the regulatory Spo-system.

It is clear that this minor exoenzyme is not involved in spore formation, as deletion of regulatory proteins had no effect on metalloproteinase production. The Spo0F protein is phosphorylated by several kinases, transferring phosphate to the Spo0B protein, then to the transcription factor Spo0A via a phosphotransfer system [21]. The Spo0K protein is involved in the formation of the competence state [22]. Spo0J protein is implicated in the process of catabolic repression of spore formation [23]. During sporulation, a set of sporulation-specific σ factors remain persistently active after septation in the following order: σH → σF → σE → σG → σK. Each sigma factor is a transcription activator of a different gene. Sigma transcription factor SigH is a key sigma factor at the sporulation initiation stage and triggers the expression of the *spo0A* gene. SigF works in the prospore during differentiation. Finally, SigK plays a crucial role in the sporulation process as the final regulator in the sigma factor cascade [24,25,26]. Metalloproteinase activity was not affected by inactivation of any of these genes.

The DegS-DegU signalling pathway primarily regulates the synthesis of MprBp metalloproteinase, a common feature among various exoenzymes [27,28,29]. It should also be noted that the minor protein is also regulated by other factors, as shown by the significant but incomplete inhibition of protein production by the deactivation of the principal regulatory proteins DegS and DegU. When the two-component DegS-DegU system fails to exert control, additional regulatory mechanisms are able to compensate and activate the metalloproteinase. This suggests that the regulation of this gene is complex, which is common to genes activated during stationary phase of growth. Similar to MprBp, the partial absence of DegS and DegU proteins preserves the activity of an extracellular subtilisin-like proteinase from the same *B. pumilus* strain 3-19 [30]. However, their total absence abolishes the expression of *B. subtilis* QB4624 subtilisin [31].

The Spo and Deg regulatory systems are globally and frequently closely interlinked. When sporulated, only a fraction of the cells in the *B. subtilis* population form endospores, resulting in a bistable state [32]. As a result, the bacterial culture separates into two distinct subpopulations-sporulating and vegetative cells. In the first group, the Spo0A transcription factor is active (Spo0A+ cells), while it remains inactive in the second group (Spo0A− cells). It has been discovered that the *degU* regulon is functionally active within the vegetative subpopulation [32]. Cells from the Spo0A+ subpopulation synthesize killer factors that cause Spo0A− cell lysis. The disrupted cell components act as a nutrient source for Spo0A+ cells, prolonging the sporulation process [33].

## 4. Materials and Methods

### 4.1. Plasmids, Bacterial Strains and Media

The strains and plasmids utilised in this study are outlined in Table 2.

Bacteria were cultured using the following media: LB medium comprising of 1.0% tryptone, 0.5% yeast extract and, 0.5% NaCl, with a pH of 8.5 [37]. The agar medium (LA) included an additional 2% agar. The media used for transformation of *B. subtilis* strains included Spitzeisen salt base medium, Spitzeisen medium I and Spitzeisen medium II [34].

Antibiotics were added to the medium for recombinant strains, with a final 20 µg/mL concentration of erythromycin for *B. subtilis* strains containing plasmid pSA1, and 20 µg/mL concentration of kanamycin for *B. subtilis* strains with DegS and DegU protein mutations. No antibiotics were added to the medium for untransformed Spo-regulatory mutants.

### 4.2. DNA Techniques

The genomic DNA of *B. pumilus* 3-19 was used to amplify the regulatory region and the *mprBp* gene sequence by polymerase chain reaction using Phusion polymerase (Thermo Scientific). mprBpDir oligonucleotide (TAACCTG**GATCC**AATCAAAGGAGGGATAGG) together with mprBpRev (CATAAA**GGATCC**CAAGCACATAGGTGTTTG) were used for the BamHI restriction site (marked in bold). The primers were selected using Vector NTI Suite 8.0 software. The amplification product, treated with BamHI restriction enzyme (NEB), was purified using the GeneJET PCR Purification Kit (Fermentas) and cloned into the plasmid pCB22, pre-treated with BglII restriction enzyme (NEB). The correct incorporation of the *mprBp* gene was subsequently validated by amplifying this gene using the flanking primers, mprBpDir and mprBpRev, followed by sequencing. The resulting recombinant plasmid was named pSA1.

*B. subtilis* cells were transformed using the protocol described by Anagnostopolous et al. [38]. DNA electrophoresis was performed in 1% agarose gel in Tris-acetate buffer (PanEco, Moscow, Russia). The 1 Kb kit from Fermentas was used for markers (Fermentas, Vilnius, Lithuania).

### 4.3. Proteolytic Activity

The metalloproteinase’s proteolytic activity was determined through azocasein hydrolysis (Sigma-Aldrich, St. Louis, MO, U.S.), according to the method outlined in [39]. The activity unit is based on the hydrolysis of 1 µg substrate per minute under experimental conditions. Productivity expressed in conventional units (c.u.) was determined by dividing the activity with the optical density gauged at a wavelength of 600 nm.

Intrinsic proteolytic activity of *B. subtilis* 8g5 strains was observed at trace levels and could be neglected when assessing metalloproteinase activity in recombinant strains.

*B. subtilis* 168 (trpC2) strains with inactivated genes of regulatory proteins of the spore formation system had their own level of proteolytic activity. Therefore, when working with these strains carrying the pSA1 plasmid, the proteolytic activity of MprBp was calculated as the difference between the level of proteolytic activity in the presence of the specific metalloproteinase inhibitor 1,10-phenanthroline (5 mM final concentration) and the level of activity without inhibitor. The presence of 1,10-phenanthroline did not affect the intrinsic proteolytic activity of non-transformed *B. subtilis* 168 (trpC2) strains.

### 4.4. Bioinformatic Analysis

Bioinformatic analysis of the promoter sequence for the *B. pumilus* 3-19 metalloproteinase was carried out using the BLAST algorithm “https://blast.ncbi.nlm.nih.gov/Blast.cgi (accessed on 13 December 2023)”, in addition to the Vector NTI Suite 8.0 program. The SignalP 6.0 algorithm “https://services.healthtech.dtu.dk/services/SignalP-6.0/ (accessed on 13 December 2023)” was employed to ascertain the probable cleavage site of the signal peptide. This particular algorithm is capable of detecting a functionally active signal peptide within an amino acid sequence. The putative regions recognised by σA for the transcription factor were detected within the promoter of the metalloproteinase gene through the use of the Softberry BPROM network server. The Vector NTI Suite 8.0 program determined potential binding sites for regulatory proteins DegU and Spo0A in the *mprBp* gene’s regulatory region.

The sequences of bacilli genome fragments freely available on the NCBI server “http://www.ncbi.nlm.nih.gov (accessed on 13 December 2023)” were used for comparative analysis: *B. pumilus* SAFR-032 (YP_001488604.1), and *B. pumilus* ATCC 7061 (ZP_03055196.1), *B. licheniformis* ATCC 14,580 (YP_081058.1).

### 4.5. Statistical Analysis

All analyses were performed at least on four biological replicates. The obtained data were processed using Statgraphics Plus 5.0. and GraphPad Prism 7.05 statistical software, and were presented as the mean ± standard deviation (SD). Student’s *t*-test analysis was used to calculate the data variance with *p* < 0.05 representing a significant difference.

## 5. Conclusions

We hypothesise that the *mprBp* gene, which is positively regulated by post-exponential regulatory systems, is associated with the DegU regulon in the vegetative subpopulation of Spo0A cells. The genes expressed in these cells are controlled by DegU regulatory proteins. We have demonstrated that metalloendopeptidase activity is partially dependent on the DegS-DegU regulatory system and independent of the Spo-system of sporulation.

## Figures and Tables

**Figure 1 ijms-25-00062-f001:**
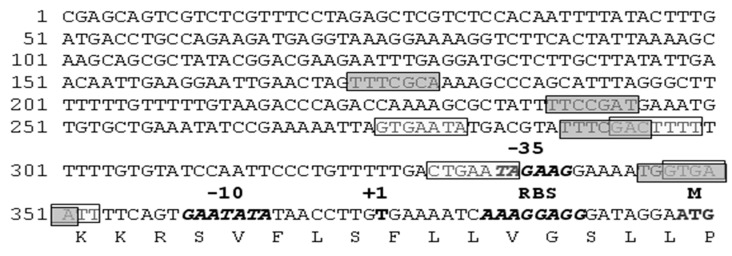
Sequence of the *B. pumilus* 3-19 *mprBp* gene promoter region. Possible sites for binding to the Spo0A protein are highlighted in dark grey, with potential binding sites for the phosphorylated form of the DegU~P protein in light grey. The −10 and −35 regions of the promoter, transcription initiation codon GTG (+1) and ribosome binding site (RBS) are depicted in bold and italics, translation initiation codon ATG is represented in bold.

**Figure 2 ijms-25-00062-f002:**
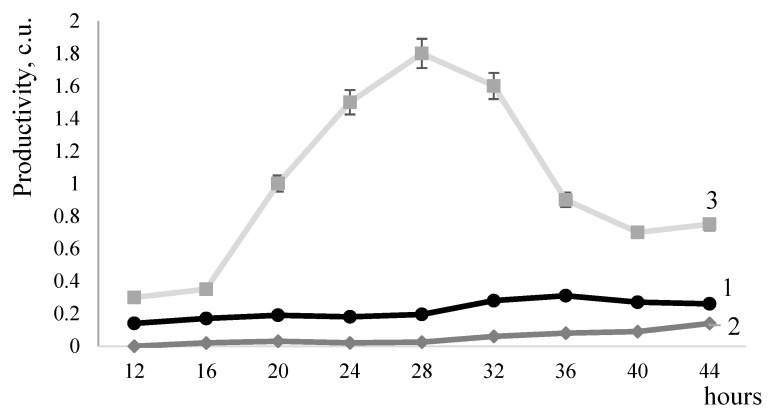
MprBp metalloproteinase productivity. 1—*B. subtilis* 8g5 pSA1 (control); 2—*B. subtilis* 8g5Δ*degS*Δ*degU* pSA1, DegS and DegU regulatory protein deficient strain, 3—*B. subtilis* 8g5 DegU32 (Hy) pSA1 with stabilization of the DegU~P protein phosphorylated form.

**Figure 3 ijms-25-00062-f003:**
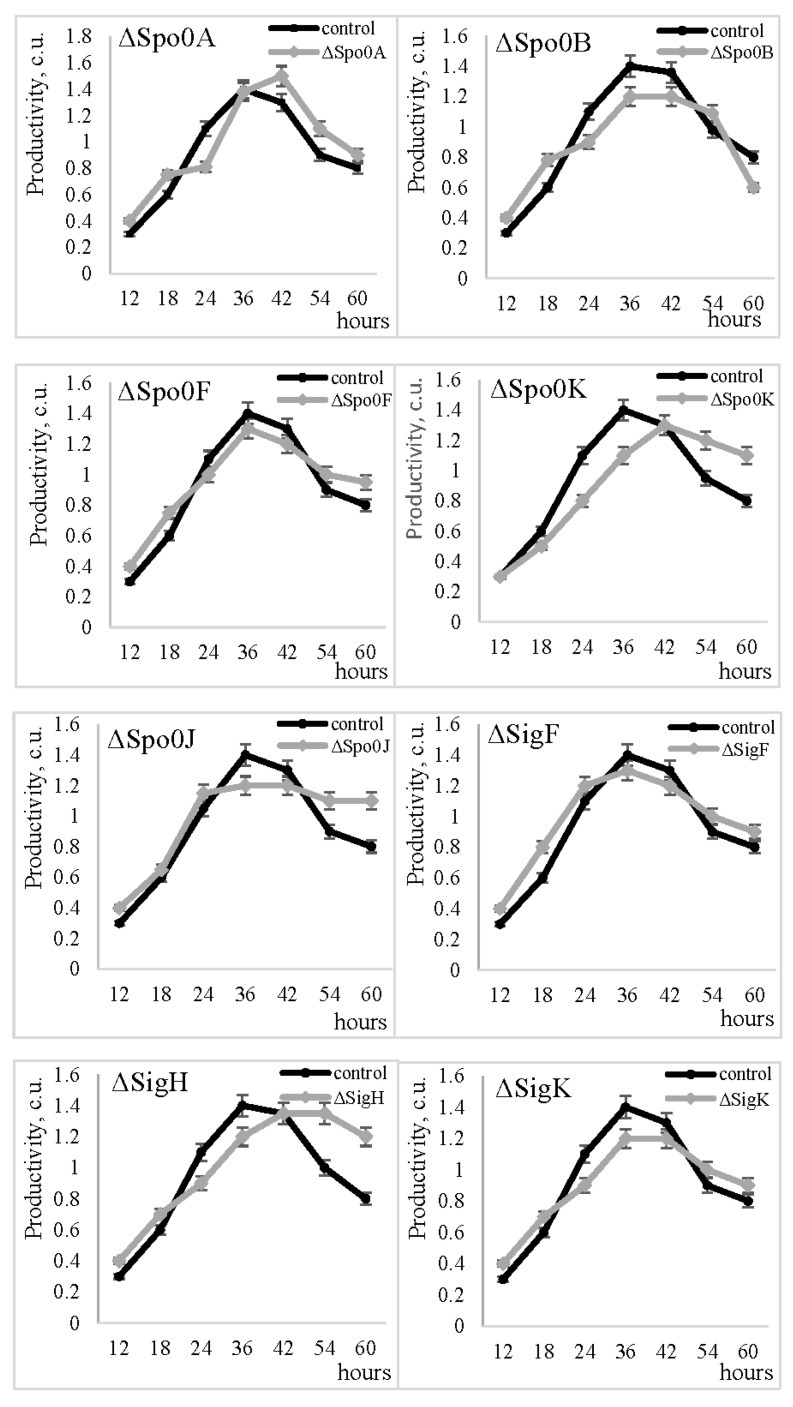
Metalloproteinase productivity in *B. subtilis* 168 (trpC2) (control) and strains defective in the Spo-regulatory proteins Spo0A, Spo0B, Spo0F, Spo0K, Spo0J, SigF, SigH, and SigK.

**Table 1 ijms-25-00062-t001:** Potential interaction sites with regulatory proteins in the *mprBp* gene promoter.

Regulatory Protein	Conservative Sequence	Number of Sites	Homology, %
Spo0A	TGTCGAA [17]	4	71–86
DegU	GTCATTAN_7_TAAATATC [18]	0	-
DegU~P	GTCATTA [18]	4	60–65

**Table 2 ijms-25-00062-t002:** Strains and plasmids used in the work.

Strains/Plasmids	Description	Source
*Bacillus pumilus* 3-19	*StrR* [1]	Laboratory collection (“Agrobioengineering” research laboratory, KFU, Kazan, Russia)
*B. subtilis* 8g5	*trpC_2_; tyr; his; nic; ura; rib; met; ade; sip* [34]	Professor Dr. Jan Maarten van Dijl University of Groningen. Department of Medical Microbiology, Netherlands
*B. subtilis* 8g5 Δ*degS*Δ*degU*	Δ*degS;* Δ*degU KmR*	
*B. subtilis* 8g5 degU32(Hy)	Hy- phenotype *KmR*	
*B. subtilis* 168 (trpC_2_)	[35]	Professor D. Zeigler, Bacillus Genetic Stock Center (BGSC), The Ohio State University, USA
*B. subtilis* 168 (trpC_2_) Δ*spo0A*	Δ*spo0A*	
*B. subtilis* 168 (trpC_2_) Δs*po0B*	Δ*spo0B*	
*B. subtilis* 168 (trpC_2_) Δ*spo0E*	Δ*spo0E*	
*B. subtilis* 168 (trpC_2_) Δ*spo0K*	Δ*spo0K*	
*B. subtilis* 168 (trpC_2_) Δ*spo0J*	Δ*spo0J*	
*B. subtilis* 168 (trpC_2_) Δ*sigF*	Δ*sigF*	
*B. subtilis* 168 (trpC_2_) Δ*sigH*	Δ*sigH*	
*B. subtilis* 168 (trpC_2_) Δ*sigK*	Δ*sigK*	
pCB22	Expression vector EU19035, *AmpR, EmR* [36]	Kostrov S.V., IMG RAS, Moscow, Russia
pSA1	With *mprBp* 1,1 kb, *EmR*	This work

## Data Availability

The data obtained in this study are available upon request.

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
