# Peer review of "Regulating Pathways of Bacillus pumilus Adamalysin-like Metalloendopeptidase Expression"

_ijms, 2023, doi:10.3390/ijms25010062_

Round 1

Reviewer 1 Report

Comments and Suggestions for Authors

This manuscript is a genetic study of the adamalysin-like peptidase MprBp of B. pumilus 3-19, a bacterial strain produced by the chemical mutagenesis of a natural strain for artificial DNase production in Russia.

   However, the target of this study is not stated, so it is unclear whether this is a basic study in protein science or whether there is a medical reason for this enzyme, which is thought to be a toxin. It should be clearly stated in the introduction. In this respect, this paper is only decdriptive about the regulation of MprBp expression and is inadequate as a scientific paper. The purpose of science is to submit a concept, while the purpose of medicine is to develop a technology with a specific goal, such as the treatment of a specific disease. 

   Although the level of genetics of this manuscript is high enough, it is too classical and inadequate as a biological study. As expected, no clear conclusion has been reached on the expression mechanism just by genetic information. Without the addition of biochemical assays, the level of modernity is inadequate. For example, the existence of several DNA sequences has been listed in this manuscript, but the basic question on them is not described. The question whether the regulatory proteins in question can bind to these sequences in vitro and in vivo has not been answered. Although it can be speculated that there may be a problem with the experimental environment, researchers should consider how to get closer to the goal in designing experiments. The attitude of simply doing what is currently possible, no matter how perfect it is, will contribute little to science and technology. This is what we leant from the development of molecular biology.

Author Response

Responses to the Reviewer's questions are given in the attached file

Reviewer 2 Report

Comments and Suggestions for Authors

Authors should pay attention to the grammar of this article

Comments on the Quality of English Language

The quality of English should be improve.

Author Response

(The authors gave the same response as above.)

Reviewer 3 Report

Comments and Suggestions for Authors

Journal: International Journal of Molecular Sciences (IJMS)

Title: Regulating pathways of Bacillus pumilus adamalysis-like metalloendopeptidase expression

Manuscript ID: ijms-2741557

In this manuscript, the role of the Deg signalling system in regulating the expression of the metalloendopeptidase gene (mprBp) of the Bacillus pumilus strain 3-19 was analysed in details, and the potential contributin of metallopeptidase enzyme (MprBp) in spore formation was investigated. MprBp is characterised as a metzincin, based on its extended motif of the active site, corresponding to the position of the third zinc ligand. Minor extracellular proteinases of bacilli may be involved in mechanisms driving the transition from free-floatng cells to biofilm formation, among others.

The mpBp metalloproteinase gene was inserted into different B. subtilis strains evidencing mutations in the genes of the DegS-DegU system regulatory proteins. Amplificationof the regulatory region and the mprBr gene sequence was carried out by using the DNA matrix of B. pumilus strain 3-19.

This study evidenced a subcontrol of mprBp gene by the DegS-DegU signalling systems. In the meantime, the gene mprBp highlighted independence from the regulatory Spo-system.

The manuscript is well written and reports an interesting aspect of bacterial adaptation, thus constituting a key for bacterial activity and adaptabilityto changed environemntal conditions. In particular adhesion and biofilm formation are pivotal aspects involved.

In the Conclusion section, addition of future research and insights suggested by the results obtained from this study could be highlighted further by the authors.

Revisions

line 2: change the name of the species to Italics;

line 77: ‘… represented in bold and underlined …’ in Figure 1, ATG is not underlined;

Figure 2: ‘Productivity, c.u.’ the first mentioning, explain c.u.;

Table 2: Uniform Italic style of the strain numbers in B. subtilis 168;

line 131: ‘… the expession of the MprBp metalloproteinase gene …’ it could be better to substitue the protein with the gene name, as follows, ‘… the expession of the mprBp metalloproteinase gene …’;

line 174: ‘B. subtilis’ change to Italic style;

line 212: ‘B. pumilus’ change to Italic style.

Author Response

(The authors gave the same response as above.)

Round 2

Reviewer 1 Report

Comments and Suggestions for Authors

Too classic to be a study of gene regulation. Biochemical evidence is needed for this type of study. 

Author Response

Dear Editor,

Thank You for your valuable feedback and suggestions. We appreciate your objective evaluation of our work and will incorporate your recommendations into our future studies.